# TGF-β Signaling Pathways in the Development of Diabetic Retinopathy

**DOI:** 10.3390/ijms25053052

**Published:** 2024-03-06

**Authors:** Andrew Callan, Sonal Jha, Laura Valdez, Lois Baldado, Andrew Tsin

**Affiliations:** School of Medicine, The University of Texas Rio Grande Valley, Edinburg, TX 78539, USA; andrew.callan01@utrgv.edu (A.C.); sonal.jha100@gmail.com (S.J.); laura.valdez01@utrgv.edu (L.V.); lois.baldado01@utrgv.edu (L.B.)

**Keywords:** diabetic retinopathy, TGF-β, transforming growth factor-β, reactive oxygen species, diabetes mellitus, retina, cell signaling, bone morphogenic proteins

## Abstract

Diabetic retinopathy (DR), a prevalent complication of diabetes mellitus affecting a significant portion of the global population, has long been viewed primarily as a microvascular disorder. However, emerging evidence suggests that it should be redefined as a neurovascular disease with multifaceted pathogenesis rooted in oxidative stress and advanced glycation end products. The transforming growth factor-β (TGF-β) signaling family has emerged as a major contributor to DR pathogenesis due to its pivotal role in retinal vascular homeostasis, endothelial cell barrier function, and pericyte differentiation. However, the precise roles of TGF-β signaling in DR remain incompletely understood, with conflicting reports on its impact in different stages of the disease. Additionally, the BMP subfamily within the TGF-β superfamily introduces further complexity, with BMPs exhibiting both pro- and anti-angiogenic properties. Furthermore, TGF-β signaling extends beyond the vascular realm, encompassing immune regulation, neuronal survival, and maintenance. The intricate interactions between TGF-β and reactive oxygen species (ROS), non-coding RNAs, and inflammatory mediators have been implicated in the pathogenesis of DR. This review delves into the complex web of signaling pathways orchestrated by the TGF-β superfamily and their involvement in DR. A comprehensive understanding of these pathways may hold the key to developing targeted therapies to halt or mitigate the progression of DR and its devastating consequences.

## 1. Introduction

Diabetic retinopathy (DR) is a complication of diabetes mellitus (DM) affecting 22% of the world’s population [1]. DR was previously understood to be a microvascular disease; however, new evidence has now shown it to be a neurovascular disease with complex pathogenesis originating from oxidative stress and advanced glycation end products [2,3,4]. The accumulation of these products leads to immune dysfunction as well as neuronal and vascular disruption and results in a disease in which multiple cell types and signaling pathways of the retina become involved [3,5,6]. Vascular dysfunction has a prime role in DR and has been studied extensively. Angiogenic pathology is found to appear early in DR as microaneurysms, small outpouchings from retinal capillaries, dot intraretinal hemorrhages, and cotton wool spots [7]. These changes can ultimately progress to vision-threatening consequences like diabetic macular edema, neovascularization, and tractional retinal detachment [4,7].

Currently, non-proliferative diabetic retinopathy (NPDR) management varies depending on the presence of macular edema [8]. NPDR without macular edema is managed by counseling patients to maintain an HbA1c below 7%. When diabetic macular edema (DME) is present, first-line treatment of intravitreal anti-vascular endothelial growth factor (anti-VEGF) therapy is started to target the vascular permeability caused by the increase in VEGF production. In cases of proliferative diabetic retinopathy (PDR), this anti-VEGF treatment is used to control neovascularization. This is often followed by pan-retinal photocoagulation laser therapy to regress new blood vessels and ablate ischemic tissue. Vitrectomies are employed in the presence of a tractional retinal detachment or persistent, large vitreous hemorrhage. Though intravitreal anti-VEGF is by far the most frequently used therapy to treat PDR and DME, it has been shown that DME patients often have a less robust response to anti-VEGF therapy compared to patients treated for neovascular age-related macular degeneration or retinal vein occlusion [9]. Serum studies show that a large percentage of DME patients have a relatively low to normal level of VEGF and therefore experience persistent, vision-limiting DME despite frequent intravitreal anti-VEGF therapy [10]. Thus, other mechanisms of DR should be explored and potentially targeted for treatment.

Current evidence suggests the TGF-β signaling family may be a potential target for the treatment of DR, as it is crucial in a wide number of homeostatic functions in the retinal vasculature, including endothelial cell barrier function and pericyte differentiation [7]. Both endothelial cells and pericytes manage the vascular environment within the retina, and thus communication between both cell types is essential for function [10]. Communication is carried out by a multitude of signaling molecules, many of which belong to the TGF-β signaling family. A study performed on mice with a conditional deletion of TGF-β found that the resulting decrease in TGF-β led to the onset of the same retinal phenotype as DR, including microaneurysms, retinal hemorrhages, and cotton wool spots [7]. This shows that disturbance of the delicate balance of these pathways is implicated in DR, as well as other similar diseases. Though changes in TGF-β expression in early DR are believed to act protectively on the retinal microvasculature, these changes over time appear to lead to endothelial cell proliferation and pericyte dedifferentiation [7,11]. Because of its role in vascular homeostasis and pathology, as well as the recent research exploring the detrimental effects caused by the dysregulation of TGF-β in the context of diabetic retinal tissues, TGF-β is proposed to be a critical factor in microvascular abnormalities caused by DM [11].

Despite the crucial role of TGF-β in homeostasis and its implication in DR pathogenesis, its precise roles remain incompletely studied [7,12]. However, current research is being conducted to identify specific TGF-β ligands associated with certain diseases, as well as the proposed mechanisms precipitating the disease state [13]. In association with receptors and intracellular signaling, the various members of the TGF-β signaling family have been shown to either diminish or enhance angiogenesis [12,14,15]. At times, previous findings contradict each other, further indicating that TGF-β signaling is not well understood in the pathogenesis of DR [12]. This necessitates further exploration of the effects of altered TGF-β signaling on the retinal vasculature. This review aims to summarize the current understanding of TGF-β signaling pathways in vascular pathology in DR. It should be emphasized that although TGF-β pathways influence numerous cell types and gene expressions in DR, these varied mechanisms are not the focus of this review and are discussed only briefly. The major pathways of DR development by the TGF-β family described in this review are illustrated in Figure 1.

## 2. An Overview of TGF-β Signaling Pathways

The TGF-β superfamily comprises more than 30 members, including TGF-β isoforms, bone morphogenetic proteins (BMPs), growth differentiation factors (GDFs), activins, inhibins, anti-Mullerian hormone, and Nodal [13,16]. They bind with several transmembrane receptor classes including activin-like kinases (ALKs), TGF-β receptors (TGFBRs), and BMP receptors (BMPRs) [13]. These ligand-receptor classes are widespread throughout various organs and tissues and are involved in embryonic development as well as homeostatic maintenance, inflammation, and various pathologies [16]. They exert fine control of a variety of tissue types in a context-dependent manner [13].

The TGF-β superfamily canonical signaling cascade begins when a ligand binds to a dimer of type II transmembrane receptors [16]. These type II receptors then recruit a dimer of type I receptors, forming a heterotetramer. Following this, type II receptors phosphorylate type I receptors, which removes the inhibitory FKBP12 protein from the type I receptor, allowing for the attachment of intracellular proteins such as SMADs. SMADs will then be phosphorylated in a cascade to form complexes and enter the nucleus to influence transcription [16,17]. There are three categories of SMADs: regulatory (R-SMAD), common mediator (co-SMAD), and inhibitory (I-SMAD). These SMADs form complexes with each other and the unique composition of SMADs affects which genes are targeted [16,18].

TGF-β can also activate a non-canonical pathway, in which receptor activation results in activation of mitogen-activated protein kinase (MAPK) pathways, IκB kinase (IKK), phosphatidylinositol-3 kinase (PI3K) and Akt, or Rho family GTPases [9,19]. These intracellular signals may then act independently of SMADs or interfere with SMADs and induce further signaling cascades which will translocate to the nucleus to influence transcription [17,18].

To help regulate these pathways, TGF-β may interact with regulatory type III receptors, betaglycan and endoglin. These receptor types consist of large extracellular components and small intracellular components with no kinase activity [16,20]. Betaglycan acts as a ligand reservoir, whereas endoglin modulates the response of ALK1 to shear stress from blood flow and thus regulates the angiogenic response [16]. Though the mechanism by which endoglin supports endothelial cell function remains unclear, studies on endoglin-null and endoglin-mutated mice have demonstrated endoglin’s crucial role in supporting this activity [20].

BMPs, a subfamily of TGF-β, appear to signal through similar canonical and non-canonical pathways to other members of the superfamily [9]. These proteins are essential in numerous developmental processes, evident in studies on different BMP knock-out mice resulting in complications like abnormal morphogenesis of various organs [21]. BMPs bind to BMP type I receptors (BMPR1) and BMP type II receptors (BMPR2) to activate the SMAD pathway or the SMAD-independent pathway. BMPR1 receptors include members of the ALK group such as ALK1, ALK2, ALK3, ALK4, and ALK6. These receptors are widely expressed in various cell types, as well as the endothelial-specific ALK-1. BMPR2 includes BMPR-II, ActRIIa, and ActRIIb and is constitutively active [9,21].

Of the numerous members of the TGF-β signaling pathway, certain ligands and receptors appear more closely related to angiogenesis in both homeostasis and pathology [16]. For example, studies on TGF-β1 knockout mice have demonstrated the role of the TGF-β1 ligand in maintaining the blood–retinal barrier. TGF-β1 is present in retinal ganglion cells (RGCs), photoreceptors, pericytes, smooth muscle cells, and microglia. Increased levels of TGF-β1 have been found in DR [16,22]. Some animal models have described that when engaging with the ALK1 receptor, TGF-β1 has pro-angiogenic effects. In contrast, the interaction between TGF-β1 and ALK5 has been shown to be anti-angiogenic [16]. ALK1 appears to be strongly expressed in embryogenesis; however, it decreases sharply in adult life until angiogenic stimulation is present. Similarly, endoglin has been shown to mitigate the activity of different TGF-β pathways to promote angiogenesis [16,23,24]. Because of this, endoglin targeting has become a potential method of inhibiting angiogenesis [24]. BMP9 and BMP10 bind with high affinity to ALK1 and endoglin to signal quiescence, depressing the angiogenic effects of VEGF [16,25]. BMP2 and BMP4 have been found to be crucial in vasculogenesis.

It should be noted that the above-described signaling model is a simplified overview. In addition to various ligands, receptors, and intracellular signaling proteins, co-receptors are also present for fine control [18]. These include betaglycan and endoglin as well as many others such as repulsive guidance molecules (RGMs), BAMBI, and crypto. Co-receptors may also have separate functions as structural proteins or ligands initiating separate signaling cascades. These co-receptors are incompletely studied and are not the focus of this review.

## 3. Association of TGF-β Signaling Pathways in DR

TGF-β signaling is an established player in maintaining retinal capillaries, and TGF-β1 has been identified as a contributing factor in the pathogenesis of DR [7,26]. TGF-β1 is known to activate in response to ROS, resulting in the eventual proliferation of endothelial cells, angiogenesis, and blood–retinal barrier disruption [27]. Inhibition of the TGF-β signaling pathway has been shown to decrease VEGF production following hypoxic states in vitro [28]. To understand the relationship between the stage of disease and TGF-β1 serum levels, one study obtained serum levels of TGF-β1 from those with NPDR and PDR diagnoses. Within these diagnosis groups, people were further divided into aflibercept treatment and non-treatment groups. Data analysis between these categories found TGF-β1 to be three times higher in patients with exacerbated PDR than those with controlled PDR and therefore predictive of disease severity and control [23]. Interestingly, patients with NPDR who had received aflibercept treatment in the past week showed lower levels of TGF-β1 than NPDR patients without aflibercept treatment. Additionally, higher TGFβ-1 levels were found to correlate with HbA1c levels, the duration of diabetes, and the progression of DR. These findings make TGF-β1 a potential predictor of disease progression from NPDR to PDR.

Other studies have shown similar findings. A study on TGF-β1 and -β2 levels in aqueous humor also found elevated levels in patients with NPDR compared to control patients [29]. Levels of TGF-β1 are also seen elevated in the vitreous humor of PDR patients compared to controls [30]. Certain polymorphisms of the TGF-β1 gene have also been studied as potential DR risk factors: it has been found that +869T/C(L10P) polymorphisms in the TGF-β1 gene may be a strong DR risk factor, whereas the 2509T/C polymorphism is not associated with DR risk [31].

TGF-β levels have also been found to correlate with the increased expression of long non-coding RNA of myocardial infarction-associated transcript (lncRNA-MIAT) [32]. LncRNA-MIAT, a known mediator in microvascular dysfunction, has been shown to be upregulated and reduce viability in adult retinal pigment epithelial cells (ARPE-19) under hyperglycemic conditions. When these cells are treated with a TGF-β inhibitor, these effects are dampened, suggesting that TGF-β may reduce the viability of epithelial cells in the setting of diabetes. The long non-coding RNA nuclear-enriched abundant transcript 1 (lncRNA-NEAT1) has also been found to trigger TGF-β1 and VEGF expression with associated findings of apoptosis and oxidative stress in diabetic mice retinas [33]. This implies that silencing the expression of lncRNA-NEAT1 could reduce the hyperglycemic stress on retinal endothelial cells.

Interestingly, the correlation between TGF-β and DR is not completely uniform. At least one study has found serum concentrations of TGF-β1 to be higher in diabetes without apparent DR than those with NPDR and PDR [34]. Likewise, studies regarding TGF-β gene expression via non-coding RNAs have revealed mixed associations. One study mirrored previous findings by measuring an increase in TGF-β related to various microRNAs in the context of proliferative DR [35]. However, in a different study utilizing the mouse model, amniotic mesenchymal stem cells migrated to hypoxic retinal tissue and reduced excessive neovascularization through the release of TGF-β1 [15]. The use of siRNA to block this pathway resulted in the negation of this effect. These results highlight that TGF-β is necessary to maintain the blood–retinal barrier, though a pathological excess expression of TGF-β may result in vasculopathy itself.

TGFβ-1 is also known to affect endothelial cell proliferation and migration, and evidence suggests that a total lack of TGF-β may also be detrimental to vascular integrity [7]. In newborn mice, total inhibition of TGFBRII signaling was shown to produce characteristics reminiscent of DR. The same study found that a lack of retinal TGF-β resulted in the dedifferentiation of microvascular pericytes, unregulated proliferation of vascular endothelium with reduced barrier function, and reactive microglia. Poor vascular function in the mouse retina led to retinal hypoxia, the induction of angiogenic molecules, and further neovascularization. This suggests that the TGF-β maintenance mechanism fails in DR and indicates that signaling pathways must be further understood if TGF-β is to be a target for future therapy.

It is possible that changes in ALK1/ALK5 expression modulate the TGF-β response in pathological conditions [11]. Using an ALK5 inhibitor, one study on rats found that the inhibition of ALK-5 signaling resulted in leaky vessels with characteristic features of DR in the embryo, newborn, and adult rate groups. The same ALK5 inhibitor was used on diabetic rats which showed prominent signs of DR. Therefore, TGF-β/ALK5 signaling is believed to be important for protection from hyperglycemic damage, especially in the setting of diabetes without DR. Similarly, the soluble expression of endoglin has been shown to promote TGF-β1/ALK-1 signaling and interfere with TGF-β1/ALK-5, therefore increasing fibro-neovascularization, angiogenesis, and arteriovenous malformations. Altogether, this activity promotes endothelial proliferation [7,36,37]. Under hyperglycemic conditions, BMP9/ALK1 signaling was shown to be disrupted in human umbilical endothelial cells, and, inversely, signaling through ALK1 solidifies the integrity of the vascular barrier by blocking VEGF-induced phosphorylation of VE-cadherin and by solidifying occludin junctions independently of VEGF [4]. These results suggest that an incremental increase in TGF-β signaling is a protective mechanism, whereas an insufficient response to TGF-β may cause disease progression [11].

Though TGF-β1 is the main isoform studied in DR, other isoforms of TGF-β have also been associated with DR [38]. One study found that the TGF-β2 isoform was the only detectable isoform in the healthy retina and did not change in concentration with DR progression, indicating that it may be an auxiliary mechanism for DR pathology [34]. A study on human retinal pigment epithelial cells showed that TGF-β2, both independently and in combination with TNF-α, is associated with retinal neovascularization and an increase in VEGF. Moreover, blockage of the TGF-β2 signaling pathway by miR-200a-3p, a microRNA shown to be downregulated in diabetic rat retinal tissue, suppressed DR progression in diabetic rats, further establishing the role of TGF-β2 overexpression in the pathogenesis of DR [39,40]. At least one study has found TGF-β3 to also be elevated in PDR [41]. Another study on diabetic rats showed elevated levels of connective tissue growth factor (CTGF), VEGF, and TGF-β2 [38]. The level of these factors was higher in more severe cases of DR. When CTGF was targeted, the levels of VEGF and TGF-β2 diminished and the apoptosis of retinal cells was reduced, providing evidence of an association between these TGF-β2 and the pathophysiology of DR.

Thus, as evidenced by mixed results from different reports in the literature, the association between TGF-β and angiogenesis in DR is complex. This is likely due to the many intracellular effectors, co-receptors, cell types, and individual gene expressions with which TGF-β interacts. As DR is an inflammatory disease, additional pathways for angiogenesis may originate in the immune response [16]. Macrophages are a significant source of TGF-β1 in DR, and TGF-β signaling is not only necessary for vascular maintenance but also for the immune response in the retina [16,30]. Studies exploring interactions between immune cells and vasculature in the environment of a pathological retina may be beneficial in elucidating these associations.

## 4. Association of BMP Signaling Pathways with DR

BMPs are a subfamily of the TGF-β superfamily and have been implicated in tumor growth, angiogenesis, and tissue and glucose homeostasis [42]. They are closely connected to vascular maintenance and survival [25]. The precise roles that BMP plays in vascular homeostasis and pathology remain unclear and are topics of continued study in DR pathogenesis [9].

BMP9 and BMP10 bind with high affinity to ALK1 via BMP type II receptors to promote vascular quiescence and stability [4,25]. Interestingly, BMP9 seems to improve vascular barrier function even in mice with total knockout of ALK1 [9]. However, BMP2, BMP4, and BMP6, through the signaling of ALK2, ALK3, and ALK6, seem to each have necessary roles in angiogenesis [16]. BMP2 is known to induce retinal endothelial barrier dysfunction by the activation of both canonical and non-canonical pathways, leading to the induction of VEGF from retinal epithelial cells and Muller cells, and the induction of angiopoietin 1, oxidative stress, and inflammatory responses [9,16]. Furthermore, this BMP2 pathway has been shown to be inhibited by BMPR inhibitors [9]. BMP2 and VEGF have also been shown to engage in positive feedback with each other at both RNA and protein levels [9,42].

Studies investigating the effects of BMP4 have found it to have both anti-angiogenic effects and pro-angiogenic effects [9,16]. To explore the BMP4 signaling mechanisms, one study analyzed BMP4 levels in ARPE cells and found that cells treated with excess glucose expressed significantly greater levels of BMP4 than those treated with a physiological amount of glucose [43]. The same study then treated ARPE cells with exogenous BMP4 and found an increase in VEGF production in a time- and dose-dependent manner. These results establish an association between elevated BMP4 levels and angiogenesis. However, this conflicts with previous studies conducted on human retinal pigment epithelial cells where there was no discernable association between BMP4 and VEGF [44]. As for BMP9, one study investigated diabetic mice with a sustained overexpression of BMP9 and found a significant decrease in retinal vascular permeability by curbing hyperglycemia-induced occludin decrease [4].

BMP has also been found to be regulated by an extracellular glycoprotein called BMP endothelial cell precursor-derived regulator (BMPER) [45]. In low concentrations, BMPER has been found to promote the migration of endothelial cells [9]. This has been demonstrated by BMPER knockout mice, showing significant endothelial permeability and, therefore, prominent vascular leakage. Conversely, high concentrations of BMPER have been found to inhibit this migration. Therefore, BMPER may be responsible for the dose-dependent fine-tuning of BMP activity. An in vitro study of human retinal endothelial cells has found BMPER to be significantly reduced in hyperglycemic conditions, which may explain BMP2 upregulation in diabetic retinas [42]. This illuminates the potential role of BMPER downregulation in the pathogenesis of DR.

## 5. Association of TGF-β with Immune Cells in DR

Both the expression of TGF-β by macrophages and the effects of TGF-β on macrophages have been implicated in the pathogenesis of DR [46]. The effects of TGF-β on macrophages have been shown to result in anti-inflammatory effects, specifically through SMAD6 and SMAD7. One pathway explored has been TGF-β’s ability to promote the action of SMAD6 to degrade MYD88, which is necessary for TLR activation of nuclear factor- κB (NF-κB). SMAD7 activation has been shown to inhibit TNF-mediated inflammatory pathways through complexing with TAB2 and TAB3 molecules [47].

These findings of TGF-β promoting anti-inflammatory effects are at odds with what has been the observed function of TGF-β expressed by macrophages, specifically in diabetic retinas. To investigate these pathways, Mondragon et al. experimented with Rhesus monkey retinal endothelial cells (RhRECs) stimulated by macrophage-derived TGF-β [48]. The results showed a significant increase in BIGH3 expression and BIGH3-mediated apoptosis of RhRECs, demonstrating the apoptotic effects of macrophage-derived TGF-β on RECs. Further demonstrating this relationship, RhRECs were treated with media conditioned by macrophages cultured under diabetic conditions, which also showed a significant increase in BIGH3 expression. When RhRECs being treated with TGF-β were cotreated with antibodies against TGF-β, BIGH3, or TGF-βRI, apoptosis was significantly blunted.

A histopathologic investigation of post-mortem retinal tissue from a donor with a 7-year history of type II DM with NPDR was conducted to see the mechanism of BIGH3 in early DM pathogenesis [46]. It was found that that the retinal arterioles had significant levels of BIGH3. Perivascular presence of microglia and macrophages were also found. This offers an in vivo representation of the role of TGF-β-induced retinal changes in the diabetic retina before proliferation takes place.

The relationship between regulatory T-cells (Tregs) and TGF-β in the pathogenesis of DR has been one of particular interest, as well. Tregs are known for their immunoregulatory activity, and, in the setting of diabetes, it has been shown that Treg levels are decreased as compared to in patients without diabetes [49,50]. The differentiation of Tregs is dependent on the transcription factor FOXP3, demonstrated by various studies on FOXP3 knockout mice resulting in multi-organ autoimmune damage [51]. FOXP3 activity is dependent on TGF-β, and in conditions of low TGF-β, Treg differentiation is notably diminished [50]. Micro-RNA (miRNA), specifically miR-155, have been shown to modulate TGF-β activity by inhibiting SMAD signaling [52]. Therefore, it has been believed that increased miR-155 may attenuate Treg activity and, consequently, its anti-inflammatory effects. To study this, Yang et al. split type II DM study participants into three groups, defined as participants with PDR, participants with NPDR, and participants with no apparent retinopathy (NAR). A control group of non-diabetic participants without retinopathy was also studied. Measurements of the participants’ TGF-β, Treg, and miR-155 levels found that the diabetic patients had a significantly lower level of Tregs, and participants with PDR had significantly lower levels of Tregs than those with NPDR, showing an inverse relationship between Treg levels and the severity of retinopathy. This relationship was also seen between TGF-β levels and the severity of retinopathy. However, levels of miR-155 showed a direct relationship with the severity of retinopathy, with PDR participants having the highest levels of miR-155 and the control group having the lowest. The results support the hypothesis that miR-155 impairs Treg and TGF-β activity in the pathogenesis of diabetic retinopathy [52].

## 6. The Role of TGF-β in DR-Related Neurodegeneration

Neurodegeneration of the retina is one of the earliest findings in diabetic retinopathy, often identifiable before vascular abnormalities are present [53]. This is a result of a myriad of cellular and molecular mechanisms, the most well-known of which include an upregulation of pro-apoptotic proteins and elevated ROS in the diabetic retinal neuronal cells. As mentioned previously, TGF-β signaling pathways are not limited to vascular control. TGF-β signaling plays important roles in immune regulation, neuronal survival, and neuronal maintenance. However, research regarding the precise mechanisms and effects of each TGF-β molecule remains inconclusive and, at times, contradictory. Experiments on RGCs under oxidative stress have been used to illustrate the mechanisms of TGF-β1 and TGF-β2 combating cellular oxidative stress [54]. These experiments demonstrate TGF-β’s ability to promote the expression of neuroprotective and antioxidative proteins like nuclear factor erythroid-2 related factor (Nfr2), Kelch-like ECH-associated protein 1 (Keap1), aldehyde dehydrogenase 3A1 (ALDH3A1), and heme oxygenase-1 (HO-1). One study found that TGF-β1 and TGF-β2 limited damage from hyperglycemia in RGCs [55]. In this study, hyperglycemic conditions increased ROS within RGCs in vitro and resulted in irreversible epigenetic changes including histone modifications, DNA methylation, and non-coding RNAs. When TGF-β1 and TGF-β2 were knocked down, RGCs proliferated less and were more sensitive to oxidative stress. This is important in the setting of diabetes because ROS is a major contributor to the cellular damage that takes place in DR. By contrast, separate studies have shown the AGE/RAGE axis seen in diabetes to increase TGF-β presence [18,56]. This has the downstream consequences of increased ROS production, suppressed antioxidant mechanisms, and increased expression of Nox enzymes.

## 7. Other Association of TGF-β Superfamily Signaling Pathway in DR

One study explored the effects of altered TGF-β2 levels on retinal cells by analyzing ARPE-19 cells treated with acrolein, a known pollutant that contributes to macular degeneration. These treated ARPE-19 cells demonstrated an increase in secreted VEGF and TGF-β2, as well as reduced viability of the treated cells. This study then repeated the treatment on a new set of ARPE-19 cells, followed by administration of a SMAD3 inhibitor. This inhibitor was intended to inhibit the effects of the increased TGF-β2. This addition saved about 48% of cells from senescence. A major mechanism of cell death may therefore be related to an increase in secreted TGF-β2 [57].

This effect may not be the same for all retina cells, however. Human retina pericytes (HRPs) have been shown to secrete increased levels of VEGF and TGF-β2 in the setting of hyperglycemia. HRP appears to increase in cell number and decrease VEGF secretion when exposed to elevated TGF-β2. A study on HRP exposed to hyperglycemic conditions showed the cells to be refractory to TGF-β2 effects compared to those in euglycemic conditions. Because elevated TGF-β2 levels have previously been shown to cause pericyte proliferation, these findings suggest that in the diabetic retina, elevated glucose levels may prevent TGF-β2 from proliferating pericytes, leaving the HRP more susceptible to damage by other mechanisms. TGF-β2 has also previously been shown to decrease VEGF levels, so reduced TGF-β2 activity allows for less VEGF regulation and, therefore, increases angiogenesis [58].

Emerging evidence has also shown that the expression of Nox enzymes and mitochondrial dysfunction engage in a positive feedback loop with TGF-β. Mitochondria produce ROS which then stimulate increased TGF-β production. This results in an increase in Nox expression in response and, therefore, a greater production of ROS. NOX-generated ROS increase mitochondrial dysfunction and further produce ROS. Redox imbalances also activate latent TGF-β1 and induce TGF-β1 gene expression. The complete blockade of TGF-β has been shown to damage ocular tissue through unregulated immune responses [18,22].

The BMP subfamily is also involved in many mechanisms outside of the vasculature with relevance to DR. BMP2 expression is induced by ROS, which in turn have been previously correlated with hyperglycemia-induced vascular calcification [59]. Dyslipidemia in diabetes has been shown to increase BMP4, leading to an increase in Nox1, as well as the induction of inflammation by Cox-2 and VCAM-1. BMP2 also releases inflammatory markers to induce adhesion of the endothelium to leukocytes and induces oxidative stress in human retinal endothelial cells (HRECs) [42,60]. BMP4 has been found to be overexpressed in Muller cells, stimulated by hyperglycemia and reduced by regulatory non-coding RNAs [60]. This is believed to contribute to DR pathogenesis.

In addition to TGF-β isoforms and BMPs, many other TGF-β family members may also be involved in DR. One such member is growth differentiating factor 15 (GDF-15), whose precise intracellular mechanisms are not yet known [61]. GDF-15 expression increases in ischemic states and has been shown to be associated with cardiovascular injury and inflammation, including retinal vascular pathologies such as DR [61,62]. In fact, previous studies have demonstrated that serum concentrations of GDF-15 can be positively correlated with DR and may hold value as biomarkers for identifying the disease [62]. GDF-15 is secreted by cells such as endothelial cells, vascular smooth muscle cells, and macrophages and may be an indication of microvascular damage. An in vitro study with human umbilical vein endothelial cells (HUVECs) showed that GDF-15 blocked angiogenesis by blocking endothelial tube formation by connective tissue growth factor 2 (CCN-2), a protein known for its proangiogenic functions [63]. GDF-15 has shown anti-inflammatory qualities by inhibiting the ability of leukocytes to interact with integrin or arrest along the vascular endothelium and by modulating pathways involved in oxidative stress [61,62]. This may be another mechanism by which GDF-15 plays a role in the pathogenesis of DR.

## 8. Other Molecular Mechanisms Implicated in DR Pathogenesis

A well-known consequence of the chronic hyperglycemia in DM is an elevated level of VEGF in the retina which promotes the aberrant angiogenesis seen in DR [64]. The specific molecular pathway implicated in this mechanism has yet to be fully elucidated. A recent study in rats found that the use of VEGF agonists promoted the protein kinase C (PKC)/endothelin (ET)/NF-kB pathway to enhance intercellular adhesion molecule 1 (ICAM-1) expression in DR [65]. ICAM-1 has been known to promote leukocyte adhesion to endothelial cells, leading to endothelial disruption and inflammation. A previous study with DM-induced rats revealed significantly decreased levels of ICAM-1 when VEGF was inhibited [65]. Therefore, this identified pathway may serve as a pharmacologic target in the treatment of DR.

Another pathway of increasing interest in the pathogenesis of DR is the Janus tyrosine kinase and signal transducers and activators of transcription (JAK/STAT) pathway [66]. A recent study explored the effects of JAK/STAT inhibition on the progression of DR and the production of VEGF in streptozosin (STZ)-induced DR model mice [66]. It was found that the STZ-induced mice had an increased activation of JAK/STAT pathways in the retina compared to the control group. Furthermore, STZ-induced mice treated with JAK inhibitors, like tofacitinib and JAK inhibitor I, showed a decreased expression of VEGF, as well as decreased retinal vascular leakage. Through immunofluorescence, it was determined that JAK1 had the greatest association with increased retinal vascular leakage compared to JAK2, JAK3, and STAT3 [66]. These results suggest that the JAK/STAT pathway may have the potential to be a therapeutic target in DR, and more research is necessary to identify the specific receptors involved in the pathogenesis of DR in humans.

## 9. Current Research on TGF-β’s Potential Role in DR Treatment

Understanding the mechanisms that the various members of the TGF-β superfamily play in the pathogenesis and progression of DR allows for research regarding potential pharmacologic intervention to prevent the onset or delay the progression of the disease. One molecule of interest is GDF11, a member of the TGF-β superfamily, which has previously been associated with the regulation of retinal neurogenesis and promotion of angiogenic activity in ischemic limb tissue in diabetic rats [67,68]. Mei et al. studied the actions of supplemental GDF11 in the retinas of diabetic rats to explore the possibility of halting the progression of DR [69]. The results of the study showed that the administration of GDF11 was protective against retinal vascular endothelial cell and retinal pericyte apoptosis, two major characteristics of DR. It was also found that pretreatment of the diabetic mice with recombinant GDF11 (rGDF11) reduced the apoptosis of retinal endothelial cells. Pretreated diabetic mice showed an increased expression of anti-apoptotic proteins like Bcl-2 and a decreased expression of pro-apoptotic proteins like Bax. Diabetic mice treated with GDF11 also displayed greater blood–retinal barrier (BRB) integrity than the non-treatment diabetic mice. Furthermore, Western blot analysis showed a greater presence of tight junction proteins in the retinas of the treatment diabetic mice, as compared to the non-treatment diabetic mice. Explorations of the possible mechanism behind these results show that GDF11 may promote the canonical TGF-β/SMAD2 pathway, as well as the noncanonical pathways of NF-κB and PI3K-Akt-FoxO.

The utility of targeting TGF-β signaling pathways for the treatment of DR has also been demonstrated in vitro using acrolein, an endogenous compound which has been previously implicated in TGF-β-mediated retinal pigment epithelium (RPE) cell death in the setting of diabetes [69,70,71]. ARPE cells incubated in glucose and acrolein showed significant cell death [71]. However, when these cells were also treated with SIS3, a specific inhibitor of SMAD3, or SB431542, a TGFβR1 antagonist, these results were almost entirely prevented. Not only do these results demonstrate a TGF-β mechanism for acrolein’s damaging effects, but they suggest the value in targeting TGF-β signaling in the treatment of DR [27,71].

## 10. Conclusions

DR is a complication of diabetes in which damage to retinal cells results in irreversible vision loss. Traditionally thought to be a vascular disease, it is now known to arise from complex and interlinked mechanisms disrupting the retinal vasculature, neurons, and immune response. The TGF-β family of proteins consists of several different proteins that participate in signaling pathways, many of which are crucial to retinal integrity and are heavily implicated in DR progression. Numerous studies have explored the molecular mechanisms of the many members of the TGF-β family. However, conflicting results indicate that it is not yet well understood on a molecular level. Current understanding of the downstream effects of TGF-β signaling is complicated by its ubiquity and the interaction between numerous cell types. This understanding is important because by expanding the available knowledge on the pathogenesis of DR, more effective pharmacological interventions can be developed to delay or prevent the onset of disease. Recent studies on targeting TGF-β signaling pathways to prevent the onset or delay the progression of DR have shown promise. Further research is needed, however, to address all various cell types with which the TGF-β proteins interact to gain a more complete understanding of DR progression at the molecular level and to develop robust pharmacologic interventions.

## Figures and Tables

**Figure 1 ijms-25-03052-f001:**
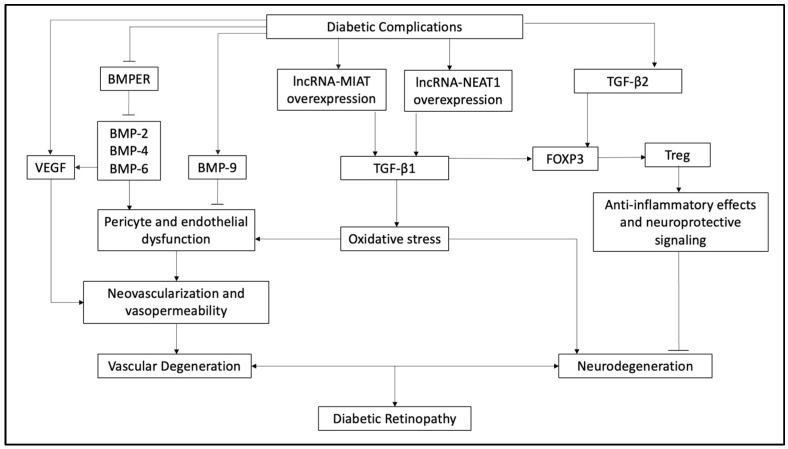
Summary of the current understanding of TGF-β roles in diabetic retinopathy. TGF-β is understood to be pervasively crucial to retinal homeostasis and is closely associated with the pathogenesis of diabetic retinopathy across multiple cell types and interactions. However, the precise action of each TGF-β superfamily member remains largely unclear. Diabetic complications encompass the myriad of destructive mechanisms that contribute to diabetic retinopathy, like hyperglycemia and immune response. Positive associations are denoted by the arrowhead (↓), and negative associations are denoted by the inhibitory arrow (I). Abbreviations are as follows: TGF = transforming growth factor; VEGF = vascular endothelial growth factor; BMP = bone morphogenic growth factor; BMPER = bone morphogenic protein endothelial receptor; FOXP3 = forkhead box protein 3; Treg = regulatory T-cell.

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
