# Peer review of "TGF-β Signaling Pathways in the Development of Diabetic Retinopathy"

_ijms, 2024, doi:10.3390/ijms25053052_

Round 1
Reviewer 1 Report
Comments and Suggestions for Authors
It's a compresive review focusing the TGF-ß signaling pathway in the development of diabetic retinopathy (DR). Discusses in detail the effects of members of the TGF protein family. Provides a summary diagram of the central role of TGFß in DR. However it would be appreciate to expand your work with a brief summary about other molecular ascpect (such as ICAM-1, JAK2/STAT3 signaling pathway and NFkappaß) to the development of diabetic retinopathy.
Author Response
Thank you for your review and helpful comments! We changed the spelling of some words and added abbreviations to the figure for additional clarification. We added information about other pathways in a new section titled "Other Molecular Mechanisms Implicated in DR Pathogenesis" towards the end of the article. We also added three additional citations for this section. The references list is up to date.
Reviewer 2 Report
Comments and Suggestions for Authors
Diabetic retinopathy (DR) is a complication of diabetes mellitus. Understanding the disease at the molecular level is important to develop a drug targeting DR. The authors suggested that the TGF-b signaling pathway has a significant role in DR pathogenesis. Although the precise mechanism remains unclear, this review summarized recent discoveries, proposing the possible connections between TGF-b signaling and diabetic retinopathy systematically. The review provides deep insight into the DR progression at the molecular level and will lead future studies on this complicated disease. I would like to recommend this manuscript be published in the International Journal of Molecular Sciences.
Author Response
Thank you for your positive review! We changed the spelling of some words and added abbreviations to the figure for additional clarification. We added information about other pathways in a new section titled "Other Molecular Mechanisms Implicated in DR Pathogenesis" towards the end of the article. We also added three additional citations for this section. The references list is up to date.